**Biological science practices**

environmental science

virtual meeting, logistics, early career, inclusivity, online accessible, questionnaire

**Author for correspondence:**
Paris V. Stefanoudis
e-mail: paris.stefanoudis@zoo.ox.ac.uk

# Moving conferences online: lessons learned from an international virtual meeting

Paris V. Stefanoudis[1,2], Leann M. Biancani[3,4], Sergio Cambronero-Solano[5,6], Malcolm R. Clark[7], Jonathan T. Copley[8], Erin Easton[9], Franziska Elmer[10], Steven H. D. Haddock[11], Santiago Herrera[12], Ilysa S. Iglesias[13], Andrea M. Quattrini[4], Julia Sigwart[14], Chris Yesson[15] and Adrian G. Glover[16]

[1]Department of Zoology, University of Oxford, Oxford, UK
[2]Nekton Foundation, Oxford, UK
[3]Department of Biology, University of Maryland, College Park, MD, USA
[4]Smithsonian Institution, National Museum of Natural History, Washington, DC, USA
[5]Departamento de Física, Universidad Nacional de Costa Rica, Heredia, Costa Rica
[6]Colectivo Internacional Pelagos Okeanos, Costa Rica
[7]National Institute of Water and Atmospheric Research, Auckland, New Zealand
[8]School of Ocean and Earth Science, University of Southampton, Southampton, UK
[9]School of Earth, Environmental and Marine Sciences, University of Texas Rio Grande Valley, Edinburg, TX, USA
[10]School for Field Studies, Center for Marine Resource Studies, Turks and Caicos Islands
[11]Monterey Bay Aquarium Research Institute, Moss Landing, CA, USA
[12]Department of Biological Sciences, Lehigh University, Bethlehem, PA, USA
[13]Department of Ocean Sciences, University of California, Santa Cruz, CA, USA
[14]School of Biological Sciences, Queen's University Belfast, Belfast, UK
[15]Zoological Society of London, London, UK
[16]Natural History Museum, London, UK

PVS, 0000-0002-4040-8364; FE, 0000-0002-3551-9510; JS, 0000-0002-3005-6246

We consider the opportunities and challenges associated with organizing a conference online, using a case study of a medium-sized (approx. 400 participants) international conference held virtually in August 2020. In addition, we present quantifiable evidence of the participants' experience using the results from an online post-conference questionnaire. Although the virtual meeting was not able to replicate the in-person experience in some aspects (e.g. less engagement between participants) the overwhelming majority of respondents found the meeting an enjoyable experience and would join similar events again. Notably, there was a strong desire for future in-person meetings to have at least some online component. Online attendance by lower-income researchers was higher compared with a past, similar-themed in-person meeting held in a high-income nation, but comparable to one held in an upper-middle-income nation. This indicates that online conferences are not a panacea for diversity and inclusivity, and that holding in-person meetings in developing economies can be at least as effective. Given that it is now relatively easy to stream contents of meetings online using low-cost methods, there are clear benefits in making all presented content accessible online, as well as organizing online networking events for those unable to attend in person.

## 1. Introduction

The global pandemic of COVID-19 and associated human-movement restrictions resulted in mass postponements or cancellations of in-person scientific meetings. Conferences are an essential part of any academic career, and offer a unique opportunity for scientists to interact, network and form partnerships.

The absence of conferences can have greater impacts on early-career researchers (e.g. students and postdoctoral graduate researchers) because they have time-sensitive academic paths that often rely on conferences to disseminate their work, connect with established researchers and identify future opportunities to advance their academic careers.

During 2020–2021 some meetings proceeded on their original timetable yet switched to fully virtual or allowed virtual attendance (online-accessible) [1]. Benefits of virtual conferences can include greater inclusivity, reduced carbon footprint and a digital archive [2–6], while challenges include fewer networking opportunities and reduced social interaction [7,8]. However, to date, there has been little quantifiable evidence on the opportunities and challenges of virtual conferences (e.g. [9]), especially in environmental sciences (e.g. [10,11]).

Here, we provide some practical considerations for prospective organizers of online conferences. We also present a case study of a recent, medium-sized (approx. 400 participants), online international meeting, provide information on logistics and demographics, and analyse participants' feedback drawn from a post-meeting questionnaire.

## 2. Pre-meeting considerations

The points below summarize general planning issues.

### (a) Is the meeting going to include only live presentations, pre-recorded or a combination?

Live presentations are more interactive, but carry more risk (e.g. presenter broadband quality), require technical support and place demands on presenters from different time zones. A pre-recorded format only depends on technical support, but is potentially less interactive. Offering both formats is more flexible but requires more planning.

### (b) Will presentations be available on demand?

This allows participants from other time zones, with a poor Internet connection, and with other duties to watch at their convenience. A logistically easy option is to use software (e.g. Zoom) that allows for live broadcasting on streaming services (e.g. YouTube) that offer automatic archiving of the streamed content by default. This content can be made publicly available or restricted to specific users as appropriate. If another platform is used that does not have the built-in capability to stream content on demand it may be logistically more complex to achieve, as it requires recording all live presentations, and uploading and hosting all presentations on the platform for the determined archival period. If presentations are made available on demand, consideration should be given to uploading standard instead of high definition videos, to minimize costs for those participants that need to buy Internet data packages. Another point is to add time stamps for the start of each presentation in the video description on YouTube, which enables the viewer to jump to specific talks by clicking on the time stamp that You-Tube automatically hyperlinks to the spot in the video. Alternatively, individual videos of each presentation can be produced and uploaded, which takes more time but also results in a much more valuable product for the presenter.

### (c) In what time zone should the meeting be held?

Identifying time zone representation of potential participants (e.g. from registration data of similar, past in-person meetings) can aid this decision. For hybrid meetings, the time zone is typically the same as the host institute, whereas for fully virtual meetings the time zone can be chosen to maximize participation and/or diversity. For international conferences, it is not possible to accommodate all participants' time zones, hence an on-demand option or splitting sessions across time zones might be appropriate (for the latter, see CarpentryCon 2020, https://2020.carpentrycon. org; GeoHab2021, http://geohab.org/geohab-2021; Virtual Island Summit 2019/2020/2021, https://islandinnovation. co/virtual-island-summit-2021).

### (d) What is the duration of the meeting?

Fewer (e.g. two to three) longer (10–12 h) days allow more individuals from different time zones to attend the meeting, although remaining focused for long periods can be challenging [12]. The alternative of more (e.g. five to seven) shorter (4–6 h) days addresses extended-session fatigue, but may exclude some people in offset time zones. Adopting a split time zone format (e.g. two sessions each day separated by 12 h) can be suitable for some international conferences, but requires additional support personnel. Large conferences often use parallel sessions for their in-person meeting and for them parallel sessions might be the only practical solution for their virtual meetings. The advantage of virtual parallel sessions is that they will likely be recorded and recordings made available, making it possible for participants to watch missed talks on demand.

### (e) Which platform(s) should you use?

Platform selection will depend on presentation format (e.g. live, pre-recorded, on demand), audience size, other services and events (e.g. networking, social, workshop) and budget. A wide variety of platforms and relevant software are available, each with their own strengths and limitations [13]. Thus, organizers are encouraged to research and actively test options thoroughly to determine which one(s) meet(s) their needs (for a discussion on the topic see [14]). Although a single platform simplifies navigation throughout the meeting for participants, it may not offer all the functionality needed for the meeting's goals. For example, having multiple platforms will often maximize digital networking between participants [15]. If more than one platform is used, however, then clear guidelines and instructions must be communicated to participants. Considerable funds can be saved by using online forms (e.g. Google forms linked to a spreadsheet) and open-source parsing programs rather than paid services. For our conference, we used python scripts (see §3a) to convert tabular spreadsheet data into conference timetables and the abstract booklet.

### (f) Are there sufficient precautions so participants feel safe interacting and presenting their data online?

Conferences are often a venue for presenting unpublished results, so it is important to take measures to ensure that information will not be recorded or posted on social media without consent. The security of platforms and whether particular platforms are banned from being used in specific sectors and/or

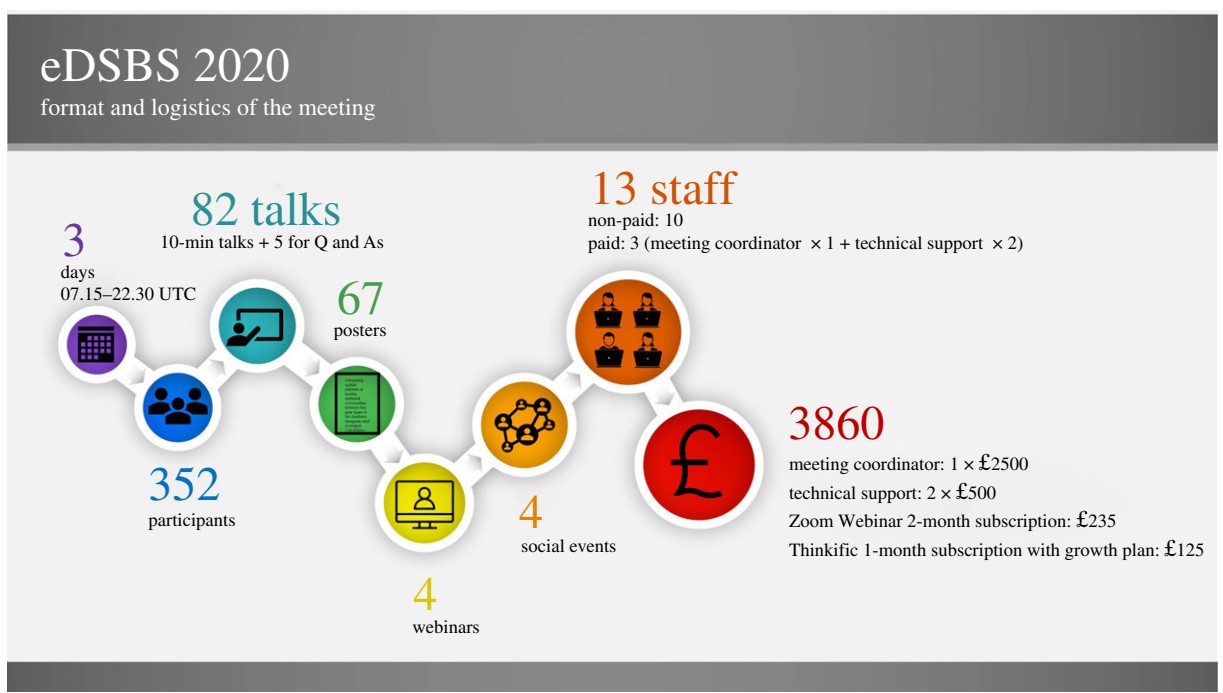

**Figure 1.** Format and logistics of eDSBS. Note that times of the meeting were selected based on the working hours of the majority of participants. Prices (in £) as of August 2020. (Online version in colour.)

countries [16] (https://support.zoom.us/hc/en-us/articles/203806119-Restricted-countries-or-regions) is important. It is essential to define a code of conduct, which should at its minimum outline examples of acceptable and unacceptable behaviours, as well as consequences for those violating the code. An example code of conduct can be found in electronic supplementary material, S1. All of the above should be clearly communicated with participants in advance so that they are confident that due diligence has been taken to make the online meeting environment a protected and safe space.

To gauge interest and aid decision-making for all these issues we recommend an informational questionnaire to be sent to likely participants (e.g. participants of previous meetings, working groups, etc.).

## 3. Case study

eDSBS was an online-only meeting held 19–21 August 2020, and organized by a UK-registered charity, the Deep-Sea Biology Society (https://dsbsoc.org/). The meeting focussed on early career researchers for whom conferences are critical career milestones, and while opportunities to present at meetings during the COVID-19 pandemic were significantly reduced.

### (a) Format and logistics

Prior to the meeting, a questionnaire was sent to the Society membership to gauge interest and help inform decisions on the meeting format (electronic supplementary material, S2). Upon feedback, the organizing committee settled on an online format that was semi-synchronous with live-streaming as well as on-demand oral and poster presentations. Figure 1 shows details of meeting format and logistics. Zoom Webinar (https://zoom.us/webinar) was used for live-streaming of oral presentations, and presenters could choose to present live or have a pre-recorded talk streamed by technical

support during their assigned session (talk-time assignments considered daytime hours of presenter's time zone). This enabled live questions and discussion similar to an in-person conference. All talks were recorded and subsequently uploaded on the main meeting website (hosted in Thinkific, https://www.thinkific.com), where they were available for 14 days after the live presentation. Therefore, regardless of participant time zone or stability of their Internet connection, participants could login to the main meeting website, open the sessions, go through the recorded talks and participate in text-based discussions in their own time. Poster presentations were not presented live, but all posters were available in poster halls on the main meeting website for the duration of the conference and the subsequent 14 days. Each poster was supported by text-based discussion and highlighted in an online poster browsing session (i.e. slide show streamed via Zoom Webinar). Finally, social events (e.g. icebreaker, early-career and student socials, closing social) were held in Zoom Webinar, where occasionally participants were randomly assigned to breakout rooms to maximize interactions and networking opportunities.

The meeting observed a strict code of conduct (electronic supplementary material, S1) to ensure participants were comfortable sharing slides and posters. Only logged-in and registered participants were able to access presentations, and it was not possible to download videos.

Numerous methods were used to communicate and share information with meeting participants. The Society website contained all essential information related to the meeting such as registration and abstract submission deadlines, schedule of talks, book of abstracts and others. Social media (e.g. Twitter) was used to advertise the event and associated deadlines, to engage a wider audience and to post scientific content during the meeting. Pre-meeting announcements, including instructions on how to access the main meeting website, links for all Zoom sessions and guidelines for presenters and session chairs, were sent via email. Slack

**Figure 2.** Breakdown of participants' responses to selected questions (Q) of the questionnaire. For the full questions see electronic supplementary material, S4. (Online version in colour.)

(https://slack.com) was used for further participant engagement, troubleshooting issues, and rapid communication of announcements and communication between organizers. Both email and Slack were used for troubleshooting and support for all the participants. Finally, Google Workspace (https://workspace.google.com/) was used to store information on logistics, data cloud, forms and to facilitate email communications. Python scripts for converting forms into abstracts and timetables are available at https://bitbucket.org/beroe/conference-generator.

## (b) Participants' feedback

The conference organizers sent a questionnaire to participants at the end of the meeting requesting participants to rate statements (scale of 1–5 corresponding to strongly disagree, disagree, neither agree nor disagree, agree, strongly agree, respectively) related to different aspects of the meeting (electronic supplementary material, S3). Participants regarded live talks as a key component of online meetings (85%; Q2, figure 2; see also electronic supplementary material, S1 table for full questionnaire results), preferred over pre-recorded talks (55%; Q24). This mirrors findings from other virtual meetings where most participants engaged

with live rather than recorded talks (e.g. [10]; P.V.S. 2021, personal observation). Participants supported the option of pre-recorded talks to cater to those with broadband issues or time zone conflicts (59–77%, Q3–Q4), but most would opt for a live talk (60%, Q25) if given the opportunity in future virtual meetings. There was no consensus on duration (38% would prefer two long days of talks without many breaks versus 42–46% would prefer 3 days with longer breaks, Q11–Q12), but participants did not favour parallel sessions (68%, Q13). Participants tended to prefer attending early in the morning (of their time zone) compared with late at night (40%), although sizable proportions were neutral (31%) or preferred the opposite (28%) (Q14). Further, presenters were overwhelmingly comfortable sharing their presentations during the meeting (91%, Q36), but half were more careful compared with in-person meetings (Q37).

Regarding engagement and networking, participants were able to connect to people whose research was of interest (71%, Q44), although the total number of questions they received and overall engagement with other researchers was deemed less than in-person meetings (55–57%, Q45–Q46). Less engagement and social interaction with other researchers has been consistently identified by organizers and participants alike as a major disadvantage of

virtual meetings (e.g. [9,17,18]). Improvements and innovations in online meeting software will probably remedy this to some extent but not completely remove the need to meet in person. There was no consensus on whether participants could concentrate as much as during in-person meetings (Q53), but the majority indicated that they could not allocate as much time as for in-person meetings (64%, Q54). Although we did not attempt to identify the sources of lack of concentration and less time committed by the participants, it most likely is the result of multiple factors such as work and personal commitments, time zone differences and online fatigue, all of which have been reported elsewhere (e.g. [10,11]). All these are likely to have been augmented due to the COVID-19 pandemic that has caused major disruption in work-life schedules for scientists of all career stages worldwide (e.g. due to care for dependants). It is also worth noting, however, that fatigue, in particular, can be as much or perhaps more during in-person meetings, due to jetlag, unfamiliar surroundings, and having to use the same facilities (e.g. sitting on lecture theatre chairs) for several hours.

Overall, the overwhelming majority of participants found the online meeting an enjoyable experience and would join similar events again (94%, Q52). Similar sentiments have been reported for online conferences either anecdotally (e.g. [19]) or by post-meeting surveys/online polls (e.g. [9,20]), although at least on one occasion when asked explicitly if online meetings are more attractive to in-person meetings the majority of respondents disagreed [11]. Interestingly, eDSBS participants indicated they want similar-themed in-person meetings in the future to have at least some online component (80%, Q56), suggesting that hybrid conferences (in person but online accessible) might become more common in the post-COVID-19 era. To this end, it is worth noting that hybrid conferences are not without their potential drawbacks: the cost is considerably more than running an in-person only meeting, and there is the risk of creating a two-tiered system favouring privileged in-person participants.

All participants' responses are available in electronic supplementary material, S4.

## (c) Organizers' feedback

Additional feedback covering logistical, technical and practical aspects of organizing a meeting is available at electronic supplementary material, S5.

## (d) Comparison with past in-person meetings

The cost of conducting the online eDSBS was only 4% of the cost of in-person meetings with similar themes and levels of attendance (figure 3a). This was achieved by the scientists themselves using 'off-the-shelf' software products and no commercial conference organization was hired. For virtual participants, the registration cost was considerably lower (approx. 90–97% less) compared with in-person meetings (figure 3b), although the costs were heavily subsidized from a grant and also those organizing the meeting did not charge for their time (as is normal for most non-profit in-person meetings as well). Overall attendance cost was even lower though, as virtual meetings do not entail travel, accommodation and subsistence costs that in-person meetings do, and which can be prohibitively expensive for many researchers.

In an effort to encourage participation and increase inclusivity, eDSBS waived registration fees for participants from developing economies and in difficult financial situations. Comparing demographic composition to a similar-themed, past in-person meeting held in the USA, a high-income country, eDSBS had almost twice as many participants from low and middle-income countries (figure 3c), and a 50% increase in the number of low and middle-income countries represented in the meeting (figure 3d). The relative numbers of participants from low and middle-income countries was a significant improvement (double) compared with the USA meeting (14% versus 7%, figure 3c). The figures from eDSBS appear to be a step in the right direction. While 14% might still seem low, deep-sea biology is in general poorly represented in low and middle-income countries [21] owing to the great cost of infrastructure for deep-sea research.

When comparing eDSBS demographics to a past in-person meeting held in Colombia—an upper-middle-income country—the number of participants from low and middle-income countries, as well as the total number of countries represented in the meeting, was comparable (figure 3c,d). Notably, the relative numbers of participants from low and middle-income countries was actually higher for the Colombia meeting (figure 3c,d), a result of increased participation of researchers from the host and neighbouring countries. This is a notable finding, suggesting online conferences are not a panacea to combat reduced participation from lower- and middle-income researchers, and that holding in-person meetings in developing economies can be at least as effective. It is also likely that if greater funds are available to bring researchers from low and middle-income countries to in-person meetings this would also help redress this balance, without needing to go online-only.

Finally, the higher representation of early-career researchers at eDSBS (figure 3c,e) was due to a combination of factors such as the meeting's particular focus on this group, the virtual format and ease of use, as well as the much lower registration costs compared with in-person meetings which has led to enhanced early-career participation elsewhere too (e.g. [5,22]). The wide participation of eDSBS from researchers of all career stages, indicates that there is an appetite among the scientific community for early-career-focused meetings with international attendance. Whereas in the past most early-career-focused meetings were regional or local, moving meetings online now allows these to take place successfully in a global space.

## 4. Concluding remarks

While it is possible to directly gain useful feedback on the successes and failures of online meetings such as the results presented here, it is not straightforward to quantify the networking and information flow in an online meeting compared with an in-person meeting. For example, it is difficult to survey all of the in-person interactions, *ad hoc* meetings, chats over coffee and casual dinner invitations, etc. that a week-long in-person conference creates, many of which may be critical to a person's career. It is important that conference organizers remain committed to providing the in-person interactions that are so essential to early-career researchers, and that the supporters of those young persons recognize the need to fund those events. At the same time,

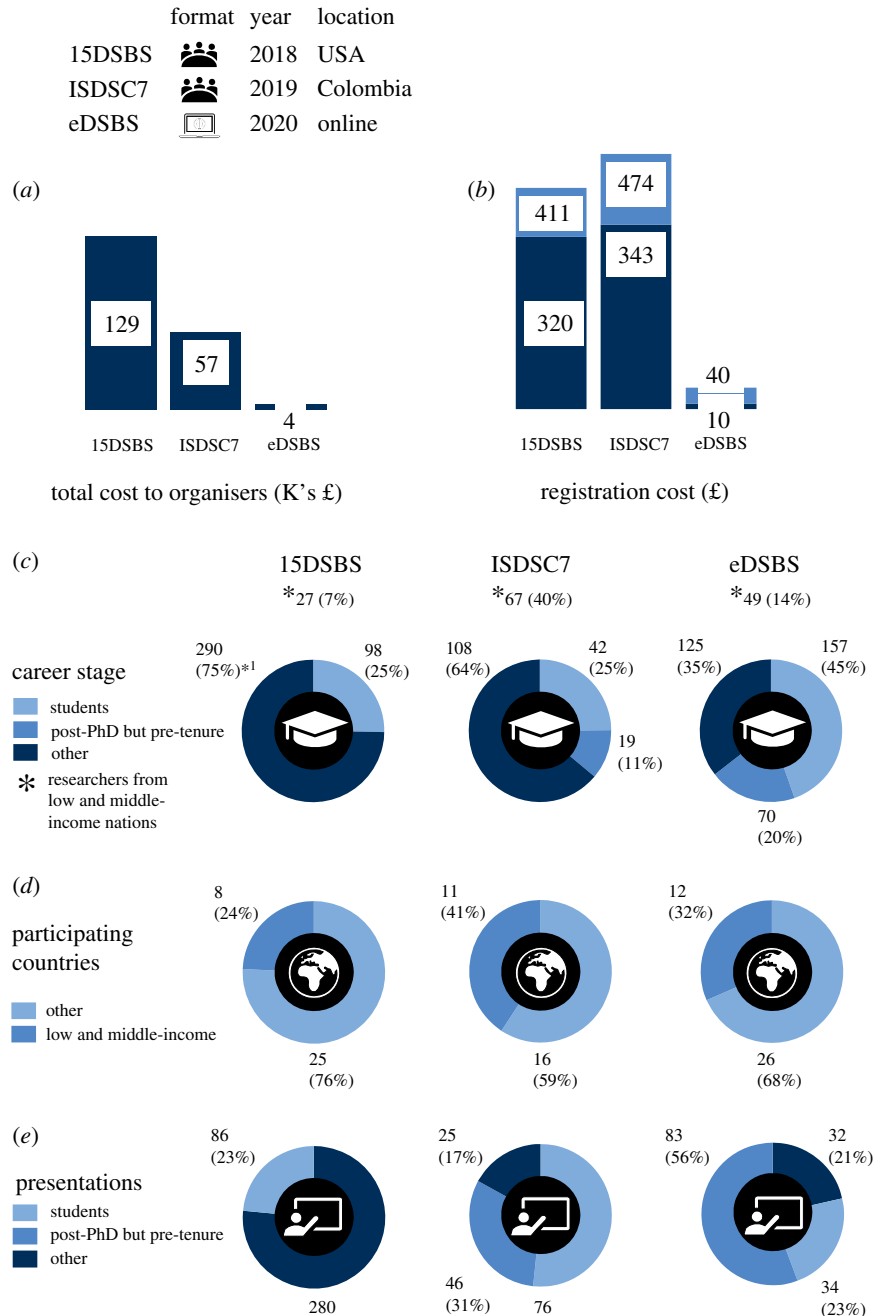

**Figure 3.** Comparison between similar-themed online and in-person meetings. (*a*) Total cost to organizers, using currency conversion rates as of 1 March 2021. Conversions were rounded to the nearest integer. (*b*) Registration cost to participants, indicating reduced (dark blue) and standard (light blue) registration fee options, linked to career stage and country of institutional affiliation. (*c*) Demographic composition by career stage. Note, that Students include PhD candidates too, while tenure includes any equivalent permanent position. (*d*) Number of participating countries, as identified from participants' institutional affiliations. Country categories based on the 2021 classification by the World Bank (last accessed on 19 January 2021). (*e*) Presentation composition by career stage. [*1]Includes post-PhD but pre-tenure. (Online version in colour.)

given the large carbon footprint that in-person meetings can carry [23], it is important not to revert fully back to the previous status quo of in-person interactions only, and embrace conference formats that are environmentally sustainable (online or online-accessible) and accessible to a broader audience. With that in mind, given that it is now relatively easy to stream all content of a meeting online, using low-cost methods, there are clear benefits, both from a moral and environmental perspective, in making all presented content at in-person meetings accessible online, as well as online networking events for those unable to attend in person.

**Data accessibility.** The datasets supporting this article have been uploaded as part of the electronic supplementary material. The data are provided in the electronic supplementary material [24].

**Authors' contributions.** P.V.S., M.C., E.E., S.H.D.H., S.H., I.S.I., A.M.Q., J.S., C.Y. and A.G.G. conceived and designed the study. P.V.S. collected and processed the data from the questionnaire survey and drafted the manuscript. All authors wrote and reviewed the manuscript, and gave final approval for publication.

**Competing interests.** The authors have no competing interest to declare.

**Funding.** This piece of research received no funding support.

**Acknowledgements.** We thank the Gordon and Betty Moore Foundation for sponsoring eDSBS, and all eDSBS participants who took the time and completed the questionnaire.

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
