## [Peer Review File · Proceedings of the Royal Society B: Biological Sciences]

Review History

RSPB-2021-0732.R0 (Original submission)

Review form: Reviewer 1

Recommendation

Accept with minor revision (please list in comments)

Scientific importance: Is the manuscript an original and important contribution to its field?

Excellent

General interest: Is the paper of sufficient general interest?

Excellent

Quality of the paper: Is the overall quality of the paper suitable?

Excellent

Is the length of the paper justified?

Yes

Should the paper be seen by a specialist statistical reviewer?

No

Do you have any concerns about statistical analyses in this paper? If so, please specify them explicitly in your report.

No

It is a condition of publication that authors make their supporting data, code and materials available - either as supplementary material or hosted in an external repository. Please rate, if applicable, the supporting data on the following criteria.

Is it accessible?

Yes

Is it clear?

Yes

Is it adequate?

Yes

Do you have any ethical concerns with this paper?

No

Comments to the Author

In this work Stefanoudis et al advocate for virtual conferences and discuss the case study of the Deep-Sea Biology Society in-person and virtual conferences. They detail features of these meetings in terms of costs, attendance numbers, duration and platforms used among others. This is a timely piece and will be a valuable addition to the growing literature examining conferences in biological sciences and other disciplines and for better and more inclusive science communication. A few of the recommendations in the piece are currently under-supported and will benefit from addition of references and would like to ask the authors to also address the following points:

1. On page 2 the authors write: "Will presentations be available on-demand? This allows participants to watch at their convenience, however, it is logistically more complex (e.g. recording of live presentations; platform to host all presentations)." – "it may be logistically more complex" would be more accurate. During small and large conferences of 2020-2021, scientific societies hosted all presentations on their platform live with recording available afterwards. Further, a number of speakers and organizers posted the presentations partially or entirely on Youtube (free to post and watch).
2. On page 2 the authors write: "In what time zone should the meeting be held? Identifying time zone representation of potential participants (e.g. from registration data of similar, past in-person meetings) can aid this decision. However, for international conferences, it might not be possible to accommodate all participants, hence, an on-demand option might be appropriate or splitting sessions across time zones." – A number of conferences (during 2020-2021 and some such as CarpentryCon for years) held their conferences in 2-3 time zones and covered all continents.
3. On page 2 the authors write: "What is the duration of the meeting? Fewer longer days allow more individuals from different time zones to attend the meeting, although remaining focused for long periods can be challenging. The alternative of more shorter days addresses the fatigue issue, but may exclude some people in different time zones. Adopting a split time-zone format (e.g. two sessions each day separated by 12 hours) can be suitable for some international conferences, but requires additional support personnel. Parallel sessions might be the only

practical solution for large, international conferences.” – In “ The alternative of more shorter”, the word “more” is redundant. Further, unlike the legacy (in-person) conferences, the parallel online sessions at virtual conferences can be recorded during virtual conferences and made available to all attendees.

4. On page 2 the authors write: “Which platform(s) should you use? Platform selection will depend on presentation format (e.g., live, pre-recorded, on-demand), audience size, other services and events (e.g., networking, social, workshop), and budget. If more than one platform is used, then clear guidelines and instructions must be communicated to participants. Considerable funds can be saved by using online forms and open-source parsing programs rather than paid services.” – For further references, a number of strategies for choosing platforms are discussed here:

a) <https://eventfund.codeforscience.org/summary-of-carpentrycon-home-session-challenges-and-opportunities-in-transitioning-meetings-online/>

b) <https://www.sciencedirect.com/science/article/pii/S1364661321000097>

It is also worth noting that more needs to be done in terms of improving the organization of virtual conferences for all attendees and some recommendations can be found here:

<https://www.nature.com/articles/s41562-021-01067-y>

5. On page 5 the authors write: “the higher representation of early-career researchers at eDSBS was a result of the meeting’s focus on this group.” – Higher early-career researcher participation at virtual conferences has been reported in a number of virtual conferences across disciplines worldwide and can also be attributed to other number factors including 1) the virtual format and ease of use and 2) the much lower registration cost (few tens of dollars compared to few hundred) for the eDSBS conference compared to the in-person version (discussed in Ref #5) and by the following reference:

<https://esajournals.onlinelibrary.wiley.com/doi/full/10.1002/bes2.1859>

7. Where fatigue is mentioned, for balance, it can be noted that in-person meeting can be as or more tiring as virtual events for the speakers and attendees (jetlag and new geographical location). Especially that physical conferences need to be held for 3-7 days with a full schedule. Further, the majority of virtual conferences were held during the time of a pandemic (2020-2021) which may also attribute to fatigue (general fatigue and fatigue from addition of daily local events moved online due to the COVID-19 pandemic).

8. One last note to authors to consider is that some or all of the data in the tables can be turned to graphs (bar plots etc) for visual impact.

Review form: Reviewer 2

Recommendation

Major revision is needed (please make suggestions in comments)

Scientific importance: Is the manuscript an original and important contribution to its field?

Acceptable

General interest: Is the paper of sufficient general interest?

Good

Quality of the paper: Is the overall quality of the paper suitable?

Acceptable

Is the length of the paper justified?

Yes

Should the paper be seen by a specialist statistical reviewer?

Yes

Do you have any concerns about statistical analyses in this paper? If so, please specify them explicitly in your report.

Yes

It is a condition of publication that authors make their supporting data, code and materials available - either as supplementary material or hosted in an external repository. Please rate, if applicable, the supporting data on the following criteria.

Is it accessible?

Yes

Is it clear?

Yes

Is it adequate?

No

Do you have any ethical concerns with this paper?

No

Comments to the Author

In the manuscript at hand, "Moving conferences online: lessons learned" submitted to Proceedings of the Royal Society B, Stefanoudis, et al. discuss the framework, as well as the positive and negative consequences of online conferences on the example of their own online conference that happened between the 19th and 21st of August, 2020. While the authors' efforts to minimise the effects of the COVID-19 pandemic on their field is laudable, the discussion here is limited to a single case study, but disguises itself as a much more general treatise. As such we would recommend a more transparent title at the very least, and a more thorough revamping of the paper at best, putting the conference in question into the context of the ongoing efforts made elsewhere, as for example the neuromatch conferences, as well as other online platforms for disseminating academic knowledge. At the current stage the paper at hand does not contribute substantially to the ongoing discussion of taking science communication online. Please find below our major and minor comments in more detail.

Major concerns:

- The title doesn't make it clear that the manuscript is a case study, not an exhaustive review. This reader finds themselves under the impression they were misled.
- The authors refer to their assessment of participants' questionnaires as quantitative, which is an overstatement in the absence of significance measures, or more far-reaching insights for more than just a single conference. The results they present here must either be assessed statistically or described as "trends". It must be made clear this is a case study.
- How does this case study differ from the other +200 case studies (found by querying Google Scholar with the terms "online conference" AND "case study" AND "covid-19") since 2021? Why would a reader read this article instead of other articles?

In summary, the manuscript at hand doesn't distinguish itself by its exhaustiveness, nor its relevance to the larger picture of moving scientific conferences online.

Minor concerns:

In the title the authors speak of 'lessons learned', though none of these are reported in the abstract. Naturally, they should be mentioned here, or eluded to.

Line 29-30: "there is little quantifiable evidence on the opportunities and challenges".

False. There are multiple case studies on conferences moving online with quantifiable assessment over their virtual proceedings as they happened. TINS' Neuromatch paper of very similar title comes to mind.

Line 43: "this allows participants to watch at their convenience, however, it is logistically more complex"

False. There are many live broadcast / teleconferencing services that offer readily available rewatch of the streamed content by default. (Youtube, Crowdcast, Zoom)

Line 54: "Parallel sessions might be the only practical solution for large, international conferences"

Parallel sessions by definition do not address the timezone discrepancy issue. What the authors might mean is "session repeats". Please clarify. By the way, the reviewers agree that timezone discrepancy is indeed one of the biggest limiting factors that need accounting for in the case of big conferences.

Line 59-60: "Considerable funds can be saved by using online forms and open-source parsing programs rather than paid services"

This needs further expansion upon the meaning of using online forms and open-source parsing programs. What do the authors mean by "open-source parsing programs"? A reference to a particular form of participant registration / content submission implementation would be helpful. A comment on how cost-saving this strategy is could be elaborated on. That's what's going to be valuable for the organisers of a future conference.

Line 62: "Consider security of platforms"

Are there any platforms less secure or vulnerable to attacks? If security is to be mentioned the authors must properly reference it. Otherwise this is just filler.

Line 62: "Define a code of conduct"

What should the code of conduct include? The paragraph could be expanded with clear examples or instructions.

Lines 112-116: "Conclusions"

The reviewers agree with the authors' reasoning and recommendation on the benefits of making all content of in-person meetings available online by default. The point should be echoed in the abstract and introduction if possible -- It is clearly one of the biggest lessons to take after surviving this pandemic and it fits with the title the authors want to use "lessons learned".

Decision letter (RSPB-2021-0732.R0)

18-May-2021

Dear Dr Stefanoudis:

I am writing to inform you that your manuscript RSPB-2021-0732 entitled "Moving conferences online: lessons learned" has, in its current form, been rejected for publication in Proceedings B.

This action has been taken on the advice of referees, who have recommended that substantial revisions are necessary. With this in mind we would be happy to consider a resubmission, provided the comments of the referees are fully addressed. However please note that this is not a provisional acceptance.

Sincerely,
Dr Maurine Neiman
<mailto:proceedingsb@royalsociety.org>

Associate Editor
Comments to Author:

Your manuscript addressing the lessons learned when moving a conference online has now been reviewed by two reviewers (of which one has collaborated with another scientist for this review). While both reviewers appreciate the timeliness of the topic, both also point out that this paper does not stand alone in addressing this topic - while you seem to claim doing so, at least with regards to the more quantitative evidence. What would thus be needed in order to consider this manuscript as a piece publishable in Proceedings B is a somewhat humbler approach regarding its significance and findings. In particular, you should aim to better place (as well as triangulate/integrate) this manuscript's findings in the light of the many other recent findings concerning the same topic. Please attend to the many useful suggestions in both reviewer's comments when doing so. In addition, I think it could be interesting to think through whether some of the table elements could better be made visual, as suggested by reviewer 1. I personally do believe that

‘quantitative’ does not necessarily imply ‘statistical’, so I would be fine with your current data approach regarding the survey

Reviewer(s)' Comments to Author:

Referee: 1

Comments to the Author(s)

In this work Stefanoudis et al advocate for virtual conferences and discuss the case study of the Deep-Sea Biology Society in-person and virtual conferences. They detail features of these meetings in terms of costs, attendance numbers, duration and platforms used among others. This is a timely piece and will be a valuable addition to the growing literature examining conferences in biological sciences and other disciplines and for better and more inclusive science communication. A few of the recommendations in the piece are currently under-supported and will benefit from addition of references and would like to ask the authors to also address the following points:

1. On page 2 the authors write: “Will presentations be available on-demand? This allows participants to watch at their convenience, however, it is logistically more complex (e.g. recording of live presentations; platform to host all presentations).” – “it may be logistically more complex” would be more accurate. During small and large conferences of 2020-2021, scientific societies hosted all presentations on their platform live with recording available afterwards. Further, a number of speakers and organizers posted the presentations partially or entirely on Youtube (free to post and watch).

2. On page 2 the authors write: “In what time zone should the meeting be held? Identifying time zone representation of potential participants (e.g. from registration data of similar, past in-person meetings) can aid this decision. However, for international conferences, it might not be possible to accommodate all participants, hence, an on-demand option might be appropriate or splitting sessions across time zones.” – A number of conferences (during 2020-2021 and some such as CarpentryCon for years) held their conferences in 2-3 time zones and covered all continents.

3. On page 2 the authors write: “What is the duration of the meeting? Fewer longer days allow more individuals from different time zones to attend the meeting, although remaining focused for long periods can be challenging. The alternative of more shorter days addresses the fatigue issue, but may exclude some people in different time zones. Adopting a split time-zone format (e.g. two sessions each day separated by 12 hours) can be suitable for some international conferences, but requires additional support personnel. Parallel sessions might be the only practical solution for large, international conferences.” – In “The alternative of more shorter”, the word “more” is redundant. Further, unlike the legacy (in-person) conferences, the parallel online sessions at virtual conferences can be recorded during virtual conferences and made available to all attendees.

4. On page 2 the authors write: “Which platform(s) should you use? Platform selection will depend on presentation format (e.g., live, pre-recorded, on-demand), audience size, other services and events (e.g., networking, social, workshop), and budget. If more than one platform is used, then clear guidelines and instructions must be communicated to participants. Considerable funds can be saved by using online forms and open-source parsing programs rather than paid services.” – For further references, a number of strategies for choosing platforms are discussed here:

a) <https://eventfund.codeforscience.org/summary-of-carpentrycon-home-session-challenges-and-opportunities-in-transitioning-meetings-online/>

b) <https://www.sciencedirect.com/science/article/pii/S1364661321000097>

It is also worth noting that more needs to be done in terms of improving the organization of virtual conferences for all attendees and some recommendations can be found here:
<https://www.nature.com/articles/s41562-021-01067-y>

5. On page 5 the authors write: “the higher representation of early-career researchers at eDSBS was a result of the meeting’s focus on this group.” – Higher early-career researcher participation at virtual conferences has been reported in a number of virtual conferences across disciplines worldwide and can also be attributed to other number factors including 1) the virtual format and ease of use and 2) the much lower registration cost (few tens of dollars compared to few hundred) for the eDSBS conference compared to the in-person version (discussed in Ref #5) and by the following reference: <https://esajournals.onlinelibrary.wiley.com/doi/full/10.1002/bes2.1859>

7. Where fatigue is mentioned, for balance, it can be noted that in-person meeting can be as or more tiring as virtual events for the speakers and attendees (jetlag and new geographical location). Especially that physical conferences need to be held for 3-7 days with a full schedule. Further, the majority of virtual conferences were held during the time of a pandemic (2020-2021) which may also attribute to fatigue (general fatigue and fatigue from addition of daily local events moved online due to the COVID-19 pandemic).

8. One last note to authors to consider is that some or all of the data in the tables can be turned to graphs (bar plots etc) for visual impact.

Referee: 2

Comments to the Author(s)

In the manuscript at hand, “Moving conferences online: lessons learned” submitted to Proceedings of the Royal Society B, Stefanoudis, et al. discuss the framework, as well as the positive and negative consequences of online conferences on the example of their own online conference that happened between the 19th and 21st of August, 2020. While the authors’ efforts to minimise the effects of the COVID-19 pandemic on their field is laudable, the discussion here is limited to a single case study, but disguises itself as a much more general treatise. As such we would recommend a more transparent title at the very least, and a more thorough revamping of the paper at best, putting the conference in question into the context of the ongoing efforts made elsewhere, as for example the neuromatch conferences, as well as other online platforms for disseminating academic knowledge. At the current stage the paper at hand does not contribute substantially to the ongoing discussion of taking science communication online. Please find below our major and minor comments in more detail.

Major concerns:

- The title doesn’t make it clear that the manuscript is a case study, not an exhaustive review. This reader finds themselves under the impression they were misled.

- The authors refer to their assessment of participants’ questionnaires as quantitative, which is an overstatement in the absence of significance measures, or more far-reaching insights for more than just a single conference. The results they present here must either be assessed statistically or described as "trends". It must be made clear this is a case study.

- How does this case study differ from the other +200 case studies (found by querying Google Scholar with the terms "online conference" AND "case study" AND "covid-19") since 2021? Why would a reader read this article instead of other articles?

In summary, the manuscript at hand doesn’t distinguish itself by its exhaustiveness, nor its relevance to the larger picture of moving scientific conferences online.

Minor concerns:

In the title the authors speak of 'lessons learned', though none of these are reported in the abstract. Naturally, they should be mentioned here, or eluded to.

Line 29-30: "there is little quantifiable evidence on the opportunities and challenges".

False. There are multiple case studies on conferences moving online with quantifiable assessment over their virtual proceedings as they happened. TINS' Neuromatch paper of very similar title comes to mind.

Line 43: "this allows participants to watch at their convenience, however, it is logistically more complex"

False. There are many live broadcast / teleconferencing services that offer readily available rewatch of the streamed content by default. (Youtube, Crowdcast, Zoom)

Line 54: "Parallel sessions might be the only practical solution for large, international conferences"

Parallel sessions by definition do not address the timezone discrepancy issue. What the authors might mean is "session repeats". Please clarify. By the way, the reviewers agree that timezone discrepancy is indeed one of the biggest limiting factors that need accounting for in the case of big conferences.

Line 59-60: "Considerable funds can be saved by using online forms and open-source parsing programs rather than paid services"

This needs further expansion upon the meaning of using online forms and open-source parsing programs. What do the authors mean by "open-source parsing programs"? A reference to a particular form of participant registration / content submission implementation would be helpful. A comment on how cost-saving this strategy is could be elaborated on. That's what's going to be valuable for the organisers of a future conference.

Line 62: "Consider security of platforms"

Are there any platforms less secure or vulnerable to attacks? If security is to be mentioned the authors must properly reference it. Otherwise this is just filler.

Line 62: "Define a code of conduct"

What should the code of conduct include? The paragraph could be expanded with clear examples or instructions.

Lines 112-116: "Conclusions"

The reviewers agree with the authors' reasoning and recommendation on the benefits of making all content of in-person meetings available online by default. The point should be echoed in the abstract and introduction if possible -- It is clearly one of the biggest lessons to take after surviving this pandemic and it fits with the title the authors want to use "lessons learned".

Author's Response to Decision Letter for (RSPB-2021-0732.R0)

See Appendix A.

RSPB-2021-1769.R0

Review form: Reviewer 1

Recommendation

Accept with minor revision (please list in comments)

Scientific importance: Is the manuscript an original and important contribution to its field?

Good

General interest: Is the paper of sufficient general interest?

Excellent

Quality of the paper: Is the overall quality of the paper suitable?

Excellent

Is the length of the paper justified?

Yes

Should the paper be seen by a specialist statistical reviewer?

No

Do you have any concerns about statistical analyses in this paper? If so, please specify them explicitly in your report.

No

It is a condition of publication that authors make their supporting data, code and materials available - either as supplementary material or hosted in an external repository. Please rate, if applicable, the supporting data on the following criteria.

Is it accessible?

Yes

Is it clear?

Yes

Is it adequate?

Yes

Do you have any ethical concerns with this paper?

No

Comments to the Author

I appreciate the time the authors have taken to address the reviewer comments. The revised manuscript is engaging and informative. I have a few final comments on the revised manuscript to authors:

1. Lines 198-199, the authors state that: "Less engagement and social interaction with other researchers has been consistently identified by organisers and participants alike as a major disadvantage of virtual meetings (e.g. 9, 17-18)." – It is important to note that the virtual conferences referenced in 9,17-18 were all held during a global pandemic and with current (first attempts) virtual organization methods and online meeting tools. So one cannot conclude that virtual conferences in general have or will always provide less engagement opportunities for

attendees. Innovations in online meeting software, formats and informed organization and planning are underway and will improve virtual conferences of 2022 and beyond. Pre- and post-conference surveys of attendees will lead the way to improvements in virtual conference formats.

2. Lines 202-207, the authors state that: "Although we did not attempt to identify the sources of lack of concentration and less time committed by the participants, it most likely is the result of multiple factors such as work and personal commitments, time zone differences, and online fatigue, all of which have been reported elsewhere (e.g. 10-11). It is worth noting, however, that fatigue in particular can be as much or perhaps more during in-person meetings, due to jetlag, unfamiliar surroundings, and having to use the same facilities (e.g. sitting on lecture theatre chairs) for several hours." –Should add that the COVID-19 pandemic has been a major disruption in work-life schedules for academics of all career stages worldwide. This likely negatively impacted experiences of virtual conference attendees in ways not compared to times of in-person conferences (pre-pandemic). For instance, the pandemic has led to additional working hours and duties for both students and faculty to create COVID-19 protocols and new classroom teaching methods and schedules. The past 18 months of the pandemic has also led to lack of childcare for scientists with parental responsibilities and resulted in full time caregiving.

3. Lines 282-283, the authors state that: "It is important that conference organisers remain committed to providing the in-person interactions that are so essential to early-career researchers, and that the supporters of those young persons recognize the need to fund those events." –It is also critical that post-pandemic conferences are only organized in environmentally sustainable (fully virtual and hybrid) formats. It is unethical and hypocritical of researchers in any discipline to spend a career claiming to perform research to improve life of humans and other living organisms only to pollute the planet with millions of tons of CO2 emissions because they wish to hold in-person scientific conferences.

References on the enormous carbon footprint of scientific conferences and how fully virtual and no flyin hybrid formats are the future of moral and environmentally sustainable scientific interactions:

<https://www.nature.com/articles/d41586-020-02057-2>

<https://www.nature.com/articles/d41586-019-03899-1>

Decision letter (RSPB-2021-1769.R0)

03-Sep-2021

Dear Dr Stefanoudis:

Your manuscript has now been peer reviewed and the reviews have been assessed by an Associate Editor. The reviewers' comments (not including confidential comments to the Editor) and the comments from the Associate Editor are included at the end of this email for your reference. As you will see, the reviewers and the Editors have raised some concerns with your manuscript and we would like to invite you to revise your manuscript to address them.

To submit your revision please log into <http://mc.manuscriptcentral.com/prsb> and enter your Author Centre, where you will find your manuscript title listed under "Manuscripts with

Decisions." Under "Actions", click on "Create a Revision". Your manuscript number has been appended to denote a revision.

Research ethics:

Use of animals and field studies:

It is a condition of publication that you make available the data and research materials supporting the results in the article (<https://royalsociety.org/journals/authors/author-guidelines/#data>). Datasets should be deposited in an appropriate publicly available repository and details of the associated accession number, link or DOI to the datasets must be included in the Data Accessibility section of the article (<https://royalsociety.org/journals/ethics-policies/data-sharing-mining/>). Reference(s) to datasets should also be included in the reference list of the article with DOIs (where available).

All supplementary materials accompanying an accepted article will be treated as in their final form. They will be published alongside the paper on the journal website and posted on the online figshare repository. Files on figshare will be made available approximately one week before the

accompanying article so that the supplementary material can be attributed a unique DOI. Please try to submit all supplementary material as a single file.

Please submit a copy of your revised paper within three weeks. If we do not hear from you within this time your manuscript will be rejected. If you are unable to meet this deadline please let us know as soon as possible, as we may be able to grant a short extension.

Best wishes,
Dr Maurine Neiman
mailto:proceedingsb@royalsociety.org

Associate Editor

Comments to Author:

You have done a good job revising your paper along the lines suggested, and sufficiently addressed all reviewers' concerns. I also appreciate your efforts to make the tables visual. In general, this worked out very well; however, I have some concerns regarding the last figure. First, please either add what ECR stands for, or perhaps rather use "PhD and postdoc"? Second, some of the text needs editing. Third, all these stars you are using make matters somewhat confusing. Is there a better way to provide this information/ is all this info necessary? Fourth, I think it would be more intuitive to have the first year (2018) on the left-hand side and the last year (2020) on the right-hand side. I was confused about this at first.

Also, please implement the last suggestions made by the reviewer. I personally would also be happy if you put the last suggestion as a question to consider, or as a trade-off. I am looking forward receiving the revised (and hopefully final!) version of your manuscript.

Reviewer(s)' Comments to Author:

Referee: 1

Comments to the Author(s).

I appreciate the time the authors have taken to address the reviewer comments. The revised manuscript is engaging and informative. I have a few final comments on the revised manuscript to authors:

1. Lines 198-199, the authors state that: "Less engagement and social interaction with other researchers has been consistently identified by organisers and participants alike as a major disadvantage of virtual meetings (e.g. 9, 17-18)." – It is important to note that the virtual conferences referenced in 9,17-18 were all held during a global pandemic and with current (first attempts) virtual organization methods and online meeting tools. So one cannot conclude that virtual conferences in general have or will always provide less engagement opportunities for attendees. Innovations in online meeting software, formats and informed organization and planning are underway and will improve virtual conferences of 2022 and beyond. Pre- and post-conference surveys of attendees will lead the way to improvements in virtual conference formats.

2. Lines 202-207, the authors state that: "Although we did not attempt to identify the sources of lack of concentration and less time committed by the participants, it most likely is the result of multiple factors such as work and personal commitments, time zone differences, and online fatigue, all of which have been reported elsewhere (e.g. 10-11). It is worth noting, however, that fatigue in particular can be as much or perhaps more during in-person meetings, due to jetlag,

unfamiliar surroundings, and having to use the same facilities (e.g. sitting on lecture theatre chairs) for several hours.” – Should add that the COVID-19 pandemic has been a major disruption in work-life schedules for academics of all career stages worldwide. This likely negatively impacted experiences of virtual conference attendees in ways not compared to times of in-person conferences (pre-pandemic). For instance, the pandemic has led to additional working hours and duties for both students and faculty to create COVID-19 protocols and new classroom teaching methods and schedules. The past 18 months of the pandemic has also led to lack of childcare for scientists with parental responsibilities and resulted in full time caregiving.

3. Lines 282-283, the authors state that: “It is important that conference organisers remain committed to providing the in-person interactions that are so essential to early-career researchers, and that the supporters of those young persons recognize the need to fund those events.” – It is also critical that post-pandemic conferences are only organized in environmentally sustainable (fully virtual and hybrid) formats. It is unethical and hypocritical of researchers in any discipline to spend a career claiming to perform research to improve life of humans and other living organisms only to pollute the planet with millions of tons of CO2 emissions because they wish to hold in-person scientific conferences.

References on the enormous carbon footprint of scientific conferences and how fully virtual and no flyin hybrid formats are the future of moral and environmentally sustainable scientific interactions:

<https://www.nature.com/articles/d41586-020-02057-2>

<https://www.nature.com/articles/d41586-019-03899-1>

Author's Response to Decision Letter for (RSPB-2021-1769.R0)

See Appendix B.

Decision letter (RSPB-2021-1769.R1)

27-Sep-2021

Dear Dr Stefanoudis

I am pleased to inform you that your manuscript RSPB-2021-1769.R1 entitled "Moving conferences online: lessons learned from an international virtual meeting" has been accepted for publication in Proceedings B, but we also suggest some minor revisions to your manuscript. Therefore, I invite you to respond to the Associate Editor's comments and revise your manuscript. Because the schedule for publication is very tight, it is a condition of publication that you submit the revised version of your manuscript within 7 days. If you do not think you will be able to meet this date please let us know.

[http://datadryad.org/submit?journalID=RSPB&manu=\(Document not available\)](http://datadryad.org/submit?journalID=RSPB&manu=(Document not available)) which will take you to your unique entry in the Dryad repository. If you have already submitted your data to dryad you can make any necessary revisions to your dataset by following the above link.

Please see <https://royalsociety.org/journals/ethics-policies/data-sharing-mining/> for more details.

Sincerely,
Dr Maurine Neiman
Editor, Proceedings B
<mailto:proceedingsb@royalsociety.org>

Associate Editor:

Board Member

Comments to Author:

I think you have overall done a fine job with your revision. But please make the following final edits:

Lines 26-28: I think this went wrong now; should be the other way around?

Line 190: This is somewhat confusing, perhaps better "not completely remove"?

Line 198: Something went wrong here; perhaps better: "e.g. due to care for dependents" or "e.g. due to lack of time because of increased care for dependents"

Figure: Replace *5 with *1 (correct in legend but not in figure itself)

Author's Response to Decision Letter for (RSPB-2021-1769.R1)

See Appendix C.

Decision letter (RSPB-2021-1769.R2)

28-Sep-2021

Dear Dr Stefanoudis

I am pleased to inform you that your manuscript entitled "Moving conferences online: lessons learned from an international virtual meeting" has been accepted for publication in Proceedings B.

Data Accessibility section

Open Access

Your article has been estimated as being 6 pages long. Our Production Office will be able to confirm the exact length at proof stage.

Paper charges

Sincerely,

Dr Maurine Neiman

mailto:proceedingsb@

Appendix A

Dear Editor,

Thank you for considering our manuscript entitled “Moving conferences online: lessons learned”.

We appreciate the time and effort put in by yourself and the two reviewers. Their constructive comments and suggestions have helped enhance the quality of our revised manuscript. Below you can find our responses to the reviewers’ comments, accompanied where necessary by line numbers so as to locate the revised text in the new version of the manuscript. Please note we have also added some additional text in the following sections (Format and Logistics; Participants’ feedback; Comparison with past in-person meetings) in order enhance the readability of the manuscript.

Note: The line numbers refer to when the document has tracked changes – show all mark up – selected.

Editor’s Comments to Author

Your manuscript addressing the lessons learned when moving a conference online has now been reviewed by two reviewers (of which one has collaborated with another scientist for this review). While both reviewers appreciate the timeliness of the topic, both also point out that this paper does not stand alone in addressing this topic – while you seem to claim doing so, at least with regards to the more quantitative evidence. What would thus be needed in order to consider this manuscript as a piece publishable in Proceedings B is a somewhat humbler approach regarding its significance and findings. In particular, you should aim to better place (as well as triangulate/ integrate) this manuscript’s findings in the light of the many other recent findings concerning the same topic. Please attend to the many useful suggestions in both reviewer’s comments when doing so. In addition, I think it could be interesting to think through whether some of the table elements could better be made visual, as suggested by reviewer 1. I personally do believe that ‘quantitative’ does not necessarily imply ‘statistical’, so I would be fine with your current data approach regarding the survey.

Reviewer 1 Comments to Author

In this work Stefanoudis et al advocate for virtual conferences and discuss the case study of the Deep-Sea Biology Society in-person and virtual conferences. They detail features of these meetings in terms of costs, attendance numbers, duration and platforms used among others. This is a timely piece and will be a valuable addition to the growing literature examining conferences in biological sciences and other disciplines and for better and more inclusive science communication. A few of the recommendations in the piece are currently unsupported and will benefit from addition of references and would like to ask the authors to also address the following points:

Comment 1: On page 2 the authors write: “Will presentations be available on-demand? This allows participants to watch at their convenience, however, it is logistically more complex (e.g. recording of live presentations; platform to host all presentations).”— “it may be logistically more complex” would be more accurate. During small and large conferences of 2020-2021, scientific societies hosted all presentations on their platform live with recording available afterwards. Further, a number of speakers and organizers posted the presentations partially or entirely on Youtube (free to post and watch).

Response: We have now reworded this section, and added additional information /suggestions regarding the use of platforms to host the presentations, in line with Comment 6 made by Reviewer 2 (**lines 67-81**).

Comment 2: On page 2 the authors write: “In what time zone should the meeting be held? Identifying time zone representation of potential participants (e.g. from registration data of similar, past in-person meetings) can aid this decision. However, for international conferences, it might not be possible to accommodate all participants, hence, an on-demand option might be appropriate or splitting sessions across time zones.”—A number of conferences (during 2020-2021 and some such as CarpentryCon for years) held their conferences in 2-3 time zones and covered all continents.

Response: Thank you for those suggestions. We have added two examples of international conferences that have used the multiple time zone format (**lines 83-88**).

Comment 3: On page 2 the authors write: “What is the duration of the meeting? Fewer longer days allow more individuals from different time zones to attend the meeting, although remaining focused for long periods can be challenging. The alternative of more shorter days addresses the fatigue issue, but may exclude some people in different time zones. Adopting a split time-zone format (e.g. two sessions each day separated by 12 hours) can be suitable for some international conferences, but requires additional support personnel. Parallel sessions might be the only practical solution for large, international conferences.”—In “The alternative of more shorter”, the word “more” is redundant. Further, unlike the legacy (in-person) conferences, the parallel online sessions at virtual conferences can be recorded during virtual conferences and made available to all attendees.

Response: The word “more” was used in order to indicate that if you choose a short-day format, then you would have to spread out the meeting over more days (e.g. a week). We have added some explanatory information in brackets to make that clear (**lines 90-93**). We have also added some text to reflect the reviewer’s point on parallel sessions can be available on demand (**lines 95-100**).

Comment 4: On page 2 the authors write: “Which platform(s) should you use? Platform selection will depend on presentation format (e.g., live, pre-recorded, on-demand), audience size, other services and events (e.g., networking, social, workshop), and budget. If more than one platform is used, then clear guidelines and instructions must be communicated to participants. Considerable funds can be saved by using online forms and open-source parsing programs rather than paid services.”— For further references, a number of strategies for choosing platforms are discussed here:

a) <https://eventfund.codeforscience.org/summary-of-carpentrycon-home-session-challenges-and-opportunities-in-transitioning-meetings-online/>

b) <https://www.sciencedirect.com/science/article/pii/S1364661321000097>

It is also worth noting that more needs to be done in terms of improving the organization of virtual conferences for all attendees and some recommendations can be found here: <https://www.nature.com/articles/s41562-021-01067-y>

Response: Thank you for the suggestion of those references. We have now reworded the text to reflect those additions (**lines 102-114**).

Comment 5: On page 5 the authors write: “the higher representation of early-career researchers at eDSBS was a result of the meeting’s focus on this group.”— Higher early-

career researcher participation at virtual conferences has been reported in a number of virtual conferences across disciplines worldwide and can also be attributed to other number factors including 1) the virtual format and ease of use and 2) the much lower registration cost (few tens of dollars compared to few hundred) for the eDSBS conference compared to the in-person version (discussed in Ref #5) and by the following reference: <https://esajournals.onlinelibrary.wiley.com/doi/full/10.1002/bes2.1859>

Response: We have now reworded the text to reflect those additional factors suggested by the reviewer and added some references as well (**lines 256-259**).

Comment 6: Where fatigue is mentioned, for balance, it can be noted that in-person meeting can be as or more tiring as virtual events for the speakers and attendees (jetlag and new geographical location). Especially that physical conferences need to be held for 3-7 days with a full schedule. Further, the majority of virtual conferences were held during the time of a pandemic (2020-2021) which may also attribute to fatigue (general fatigue and fatigue from addition of daily local events moved online due to the COVID-19 pandemic).

Response: For balance, we have added some text (**lines 201-207**) after term “online fatigue” is mentioned.

Comment 7: One last note to authors to consider is that some or all of the data in the tables can be turned to graphs (bar plots etc) for visual impact.

Response: We have now turned previous tables into figures / infographics. Specifically, previous Table 1 is now Figure 1, and previous Table 2 is now Figure 3. We have also added an additional Figure (Figure 2) which provides a quick visual summary of the questionnaire results that are being discussed in the “Participants’ feedback” session. We believe that with those changes the visual impact of the information contained in the previous tables has increased significantly.

Reviewer 2 Comments to Author

In the manuscript at hand, “Moving conferences online: lessons learned” submitted to Proceedings of the Royal Society B, Stefanoudis, et al. discuss the framework, as well as the positive and negative consequences of online conferences on the example of their own online conference that happened between the 19th and 21st of August, 2020. While the authors’ efforts to minimise the effects of the COVID-19 pandemic on their field is laudable, the discussion here is limited to a single case study, but disguises itself as a much more general treatise. As such we would recommend a more transparent title at the very least, and a more thorough revamping of the paper at best, putting the conference in question into the context of the ongoing efforts made elsewhere, as for example the neuromatch conferences, as well as other online platforms for disseminating academic knowledge. At the current stage the paper at hand does not contribute substantially to the ongoing discussion of taking science communication online. Please find below our major and minor comments in more detail.

Major concerns

Comment 1: The title doesn’t make it clear that the manuscript is a case study, not an exhaustive review. This reader finds themselves under the impression they were misled.

Response: We have now reworded the title to reflect the case study on which this work is based upon (**lines 1-2**); similar titles have been used in similar literature (e.g. <https://dl.acm.org/doi/abs/10.1145/3456859.3456866>;

<https://rupress.org/jem/article/217/9/e20201467/151994/Lessons-of-COVID-19-Virtual-conferencesLessons-of>).

Comment 2: The authors refer to their assessment of participants' questionnaires as quantitative, which is an overstatement in the absence of significance measures, or more far-reaching insights for more than just a single conference. The results they present here must either be assessed statistically or described as "trends". It must be made clear this is a case study.

Response: We appreciate the reviewers' comment, however, to our view quantitative does not necessarily imply statistical. We only used the word once in the abstract, and have now replaced the word quantitative with quantifiable (**line 25**). We also mention in the abstract that the results are based on a case study, which in combination with the revised title should make it clear to readers that is not a general treatise / review of lessons learned from moving conferences online (**lines 20-22**).

Comment 3: How does this case study differ from the other +200 case studies (found by querying Google Scholar with the terms "online conference" AND "case study" AND "covid-19") since 2021? Why would a reader read this article instead of other articles?

In summary, the manuscript at hand doesn't distinguish itself by its exhaustiveness, nor its relevance to the larger picture of moving scientific conferences online.

Response: While we agree with the reviewers that there are several articles / case studies that deal with online conferences and discuss their advantages and disadvantages and / or proceed to make some recommendations, we do not think their number of peer-reviewed or quantifiable articles is as high as the reviewers suggest. The majority of papers that have been published over the last 2 years and that we are aware of, do not provide quantifiable evidence (i.e. in the form of a questionnaire / survey) of the participants' experience, and are based mostly on qualitative information (general impressions, personal beliefs, informal conversations etc.). We believe this is one of the key contributions of our paper, because having that knowledge will provide examples to future meeting organisers of what does work well in online meetings, and what less so.

We used the terms suggested by the reviewers in Google Scholar, and there were 573 results between 2020-2021 (as of 2 August 2021); however, after inspecting the top 30 results, only 5 actually dealt with online conferences, while the rest mentioned that phrase somewhere in the manuscript. In addition, from those top 30 results, most concerned higher education and online learning rather than academic conferences.

We also performed searches using similar keywords in Scopus. We acknowledge despite Scopus being one of the largest abstract and citation databases of peer-reviewed literature, it will still miss out part of the available bibliography. However, it provides a good indication of trends. Restricting the keyword searches in the title and abstract only (performing the search across the whole manuscript text, returned >300 results but with a lot of false positives, similar to the Google Scholar search above) the number of studies concerning online conferences and using questionnaires / surveys was <30 (as of 2 August 2021; see search link here: <https://tinyurl.com/5e3bard4>, however, note it requires having an account to Scopus to be able to access it). From those the majority concerned the field of neuroscience and biomedicine, while only one was in the field of ecology (Raby & Madden, 2021, <https://doi.org/10.1002/ece3.7251>). As a result, we believe our manuscript provides some valuable lessons learned, and is of relevance to the community the journal.

Minor concerns:

Comment 4: In the title the authors speak of 'lessons learned', though none of these are reported in the abstract. Naturally, they should be mentioned here, or eluded to.

Response: We have now modified the abstract extensively to reflect some of the key take-away messages (**lines 18-37**).

Comment 5: Line 29-30: "there is little quantifiable evidence on the opportunities and challenges".

False. There are multiple case studies on conferences moving online with quantifiable assessment over their virtual proceedings as they happened. TINS' Neuromatch paper of very similar title comes to mind.

Response: Please see response to comment 3. We have now modified the text to make it more specific, and indicate that there has been little evidence for environmental science conferences (**lines 51-52**).

Comment 6: Line 43: "this allows participants to watch at their convenience, however, it is logistically more complex"

False. There are many live broadcast / teleconferencing services that offer readily available rewatch of the streamed content by default. (Youtube, Crowdcast, Zoom).

Response: Often there may be the additional requirement to have time-limited access to these videos, and to limit access to the registrants, which can increase the logistical complexity. We do however agree with the reviewers that those options should be mentioned, and have modified the text accordingly (**lines 67-81**). Finally, it is worth noting, that the platforms mentioned do not always separate each talk individually, and contain all talks from one session in one, which is not as valuable for participants interested to watch a particular talk. We have added a comment on having time stamps on YouTube to help overcome that (**lines 76-81**).

Comment 7: Line 54: "Parallel sessions might be the only practical solution for large, international conferences"

Parallel sessions by definition do not address the time zone discrepancy issue. What the authors might mean is "session repeats". Please clarify. By the way, the reviewers agree that time zone discrepancy is indeed one of the biggest limiting factors that need accounting for in the case of big conferences.

Response: We agree that session repeats are another way of addressing the time zone issue without participants having to miss talks. We have now reworded the section "In what time zone should the meeting be held?", which also reflects a similar comment from reviewer 1 (**lines 95-100**). We have the parallel sessions comment on the section "What is the duration of the meeting?" because in the case of a large international conference with many presentations, holding repeat sessions might not be practical. However, we added text saying that and if the meeting is recorded these parallel sessions can then be available for anyone to watch on-demand (**lines 98-100**).

Comment 8: Line 59-60: "Considerable funds can be saved by using online forms and open-source parsing programs rather than paid services"

This needs further expansion upon the meaning of using online forms and open-source parsing programs. What do the authors mean by "open-source parsing programs"? A reference to a particular form of participant registration / content submission implementation

would be helpful. A comment on how cost-saving this strategy is could be elaborated on. That's what's going to be valuable for the organisers of a future conference.

Response: We have now added text to provide some concrete examples (**lines 111-114; 172-173**).

Comment 9: Line 62: "Consider security of platforms"

Are there any platforms less secure or vulnerable to attacks? If security is to be mentioned the authors must properly reference it. Otherwise this is just filler.

Response: We added text to reflect the fact there have been privacy concerns with some software / platforms used (e.g. Zoom), which has led to some government employees (e.g. Germany, USA) not being able to use them (Aiken, 2020, <https://doi.org/10.1177/0306422020935792>). In addition, Zoom is unavailable from specific countries due to regulatory reasons (<https://support.zoom.us/hc/en-us/articles/203806119-Restricted-countries-or-regions>) (**lines 120-122**).

Comment 10: Line 62: "Define a code of conduct"

What should the code of conduct include? The paragraph could be expanded with clear examples or instructions.

Response: This has now been done and we reference an example code of conduct that is available in the supplementary material (**lines 122-127; Appendix 2**).

Comment 11: Lines 112-116: "Conclusions"

The reviewers agree with the authors' reasoning and recommendation on the benefits of making all content of in-person meetings available online by default. The point should be echoed in the abstract and introduction if possible -- It is clearly one of the biggest lessons to take after surviving this pandemic and it fits with the title the authors want to use "lessons learned".

Response: We have now reworded the abstract to reflect this important point, and added some additional text in the conclusions section (**lines 34-37; 283-286**).

Appendix B

Dear Editor,

Thank you for considering our manuscript entitled “Moving conferences online: lessons learned from an international virtual meeting”.

We appreciate the time and effort put in by yourself and the reviewer. The constructive comments and suggestions have helped enhance the quality of our revised manuscript. Below you can find our responses to yours and the reviewer’s comments, accompanied where necessary by line numbers so as to locate the revised text in the new version of the manuscript.

Note: The line numbers refer to when the document has tracked changes – show all mark up – selected.

Editor’s Comments to Author

You have done a good job revising your paper along the lines suggested, and sufficiently addressed all reviewers' concerns. I also appreciate your efforts to make the tables visual. In general, this worked out very well; however, I have some concerns regarding the last figure.

Comment 1: First, please either add what ECR stands for, or perhaps rather use "PhD and postdoc"?

Response: We have now replaced ECR with Post-PhD but pre-tenure or equivalent permanent position. Note, that PhD students are included are Students, and this was clarified in the figure legend (**lines 268-270**).

Comments 2-3: Second, some of the text needs editing. Third, all these stars you are using make matters somewhat confusing. Is there a better way to provide this information/ is all this info necessary?

Response: We have now removed all numbered asterisks but for one, and simplified the text in the Figure legend (**lines 265-272**).

Comment 4: Fourth, I think it would be more intuitive to have the first year (2018) on the left-hand side and the last year (2020) on the right-hand side. I was confused about this at first.

Response: We have changed that and present results across the figure from the 2018 to 2020.

Also, please implement the last suggestions made by the reviewer. I personally would also be happy if you put the last suggestion as a question to consider, or as a trade-off. I am looking forward receiving the revised (and hopefully final!) version of your manuscript.

Reviewer 1 Comments to Author

I appreciate the time the authors have taken to address the reviewer comments. The revised manuscript is engaging and informative. I have a few final comments on the revised manuscript to authors:

Comment 1: Lines 198-199, the authors state that: “Less engagement and social interaction with other researchers has been consistently identified by organisers and participants alike as a major disadvantage of virtual meetings (e.g. 9, 17-18).”—It is important to note that the virtual conferences referenced in 9,17-18 were all held during a global pandemic and with current (first attempts) virtual organization methods and online meeting tools. So one cannot conclude that virtual conferences in general have or will always provide less engagement opportunities for attendees. Innovations in online meeting software, formats and informed organization and planning are underway and will improve virtual conferences of 2022 and beyond. Pre- and post-conference surveys of attendees will lead the way to improvements in virtual conference formats.

Response: We have changed the text (**lines 189-190**).

Comment 2: Lines 202-207, the authors state that: “Although we did not attempt to identify the sources of lack of concentration and less time committed by the participants, it most likely is the result of multiple factors such as work and personal commitments, time zone differences, and online fatigue, all of which have been reported elsewhere (e.g. 10-11). It is worth noting, however, that fatigue in particular can be as much or perhaps more during in-person meetings, due to jetlag, unfamiliar surroundings, and having to use the same facilities (e.g. sitting on lecture theatre chairs) for several hours.”—Should add that the COVID-19 pandemic has been a major disruption in work-life schedules for academics of all career stages worldwide. This likely negatively impacted experiences of virtual conference attendees in ways not compared to times of in-person conferences (pre-pandemic). For instance, the pandemic has led to additional working hours and duties for both students and faculty to create COVID-19 protocols and new classroom teaching methods and schedules. The past 18 months of the pandemic has also led to lack of childcare for scientists with parental responsibilities and resulted in full time caregiving.

Response: This has now been addressed (**lines 196-198**).

Comment 3: Lines 282-283, the authors state that: “It is important that conference organisers remain committed to providing the in-person interactions that are so essential to early-career researchers, and that the supporters of those young persons recognize the need to fund those events.”—It is also critical that post-pandemic conferences are only organized in environmentally sustainable (fully virtual and hybrid) formats. It is unethical and hypocritical of researchers in any discipline to spend a career claiming to perform research to improve life of humans and other living organisms only to pollute the planet with millions of tons of CO2 emissions because they wish to hold in-person scientific conferences.

References on the enormous carbon footprint of scientific conferences and how fully virtual and no flyin hybrid formats are the future of moral and environmentally sustainable scientific interactions:

<https://www.nature.com/articles/d41586-020-02057-2>

<https://www.nature.com/articles/d41586-019-03899-1>

Response: We have now added some text to reflect that (**lines 287-291**).

Appendix C

Dear Editor,

Thank accepting our manuscript “Moving conferences online: lessons learned from an international virtual meeting”.

Below you can find our responses to your comments accompanied where necessary by line numbers so as to locate the revised text in the new version of the manuscript. Please note we have now removed the figures from the main manuscript and provide the figure captions at the end document.

Editor’s Comments to Author

I think you have overall done a fine job with your revision. But please make the following final edits:

Comment 1

Lines 26-28: I think this went wrong now; should be the other way around?

Response: Apologies for the confusion (on our part) with this. This has now been reworded (lines 25-27).

Comment 2

Line 190: This is somewhat confusing, perhaps better “not completely remove”?

Response: This has now been changed (line 186).

Comment 3

Line 198: Something went wrong here; perhaps better: “e.g. due to care for dependents” or “e.g. due to lack of time because of increased care for dependents”

Response: This has now been changed (line 194).

Comment 4

Figure: Replace *5 with *1 (correct in legend but not in figure itself).

Response: Done; please see new version of the figure uploaded as a separate file (Figure 3_HighRes.pdf)